# GROUNDING MULTIMODAL LARGE LANGUAGE MODEL IN GUI WORLD

**Weixian Lei, Difei Gao, Mike Zheng Shou**✉
Show Lab, National University of Singapore

## ABSTRACT

Recent advancements in Multimodal Large Language Models (MLLMs) have accelerated the development of Graphical User Interface (GUI) agents capable of automating complex tasks across digital platforms. However, precise GUI element grounding remains a key challenge for accurate interaction and generalization. In this work, we present an effective GUI grounding framework, which includes an automated data collection engine that gathers extensive GUI screenshots and annotations to ensure broad generalization. We also propose a lightweight and flexible GUI grounding module designed to efficiently localize UI elements by pre-training on the collected data, and introduce a novel method to integrate this module with MLLMs for the effective execution of GUI tasks. Our approach demonstrates superior performance in task accuracy and adaptability, as validated by benchmarks such as ScreenSpot, MiniWob, AITW, and Mind2Web. Our code and data are released at https://github.com/showlab/AssistGUIGround.

## 1 INTRODUCTION

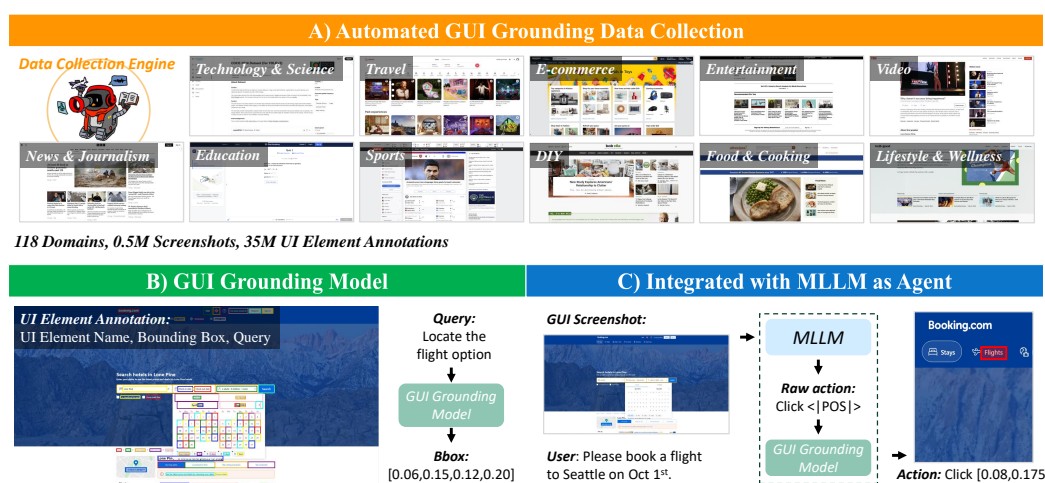

Figure 1: **Overview of our pipeline for building a vision-based GUI agent. (A)** A data collection engine gathers a diverse set of website screenshots spanning a broad range of topics. **(B)** A GUI grounding model trained on the collected data with dense elements and annotations. **(C)** The grounding model is integrated with MLLMs to form a vision-based agent.

Recent advancements in Large Language Models (LLMs) (Achiam et al., 2023; Touvron et al., 2023a;b; Jiang et al., 2023; Chiang et al., 2023) and Multimodal Large Language Models (MLLMs) (Li et al., 2023; Zhu et al., 2023; Liu et al., 2024; Bavishi et al., 2023; Hong et al., 2023) have significantly improved multimedia comprehension, logical reasoning, and decision-making capabilities, driving the evolution of Graphical User Interface (GUI) agents. These agents are designed to automate complex tasks on digital platforms such as PCs and mobile devices, reducing human effort in performing monotonous duties (Cheng et al., 2024; Gao et al., 2024; Shi et al., 2017; Zhou et al., 2023; Koh et al., 2024; Deng et al., 2023; Chen et al., 2024a;b; Lu et al., 2024a; Yan et al., 2023; Wang et al., 2024b; Yang et al., 2023b; Shaw et al., 2023; Lin et al., 2024).

✉ Corresponding Author.

Some pioneering works (Deng et al., 2023; Gur et al., 2023; Zheng et al., 2023; Kim et al., 2024) have designed text-only agents that feed structured plain text representations of screen content, such as HTML and DOM, to LLMs for reasoning, planning, and executing specific GUI actions. However, plain text representations of GUIs have inherent limitations. First, structured text is not always readily available, restricting its use. Second, raw text metadata—often extensive and noisy—requires substantial preprocessing and filtering, which increases computational latency and inference costs and may compromise model accuracy.

Humans interact with digital interfaces by visually perceiving content, making decisions, and executing actions accordingly. Drawing inspiration from this natural interaction with GUIs, another approach adopted by GUI agents is to use visual renderings as inputs. This approach circumvents the limitations associated with text metadata processing and is better suited for generalization across various platforms. Several pretrained MLLMs (Bavishi et al., 2023; Hong et al., 2023) included UI data during pre-training and have demonstrated their ability to understand basic elements within UI screenshots. A recent work, SeeClick (Cheng et al., 2024), trained a visual GUI agent using screenshots from web and mobile interfaces, enhancing vision-based capabilities in GUI tasks. Despite these advancements, their performance in GUI automation tasks still falls short compared to text-based models.

We identify a primary bottleneck is GUI element grounding, which involves aligning the LLM-generated textual plan with precise screen locations. Robust GUI agents require strong grounding capabilities, characterized by: 1) Precision in Targeting and Interaction: The agent must accurately identify and interact with the necessary elements at each step of a task, ensuring the reliable execution of complex tasks. 2) Generalization Across GUI Scenarios: The agent should effectively generalize across diverse real-world scenarios, such as online shopping, entertainment, and software-based office tasks. 3) Flexible Grounding Adaptation: The grounding model should seamlessly integrate with any LLM planner, enabling easy invocation and adaptation for planning and decision-making.

The main challenge of building such a system is developing this capability in the absence of large-scale annotations available on the internet, similar to a relevant topic, segmentation (Kirillov et al., 2023a). Therefore, a carefully tailored approach to data curation and methodical model training is essential. From the data curation, the grounding model should be trained on a comprehensive and diverse set of GUI screenshots and annotations to ensure strong generalization to various GUI applications. Currently, public datasets lack dense, diverse, and richly annotated GUI grounding data. To resolve this, we have created an automated GUI data collection engine to gather relevant screenshots and grounding annotations, as shown in Figure. 1(A). We also perform data cleaning to maintain high quality and augment query expression for different GUI elements to enhance generalizability.

From a model perspective, our goal is to develop a lightweight yet powerful grounding model and seamlessly integrate it into an MLLM to perform complex GUI tasks, as shown in Figure. 1(B) & (C). Therefore, we first design a GUI Grounding model with a robust image encoder for generating embeddings and a language encoder for processing queries, combined in a lightweight decoder to predict bounding boxes. This well-trained model is then integrated with the MLLM for downstream tasks like navigation. We add a `<|POS|>` token to the MLLM's vocabulary, which triggers grounding when needed. Its hidden embedding is passed to the GUI grounding model to locate the corresponding position for operation. This design separates reasoning and grounding into modular components, utilizing the MLLM for reasoning and the grounding model for precise localization.

We evaluated our approach on the ScreenSpot (Cheng et al., 2024) benchmark, demonstrating the efficacy of our GUI grounding pre-training methods. Additionally, we integrated our model with MLLMs to function as a comprehensive GUI agent. This agent has been adapted for various mobile and web-based tasks, including MiniWob (Shi et al., 2017), AITW (Rawles et al., 2024), and Mind2Web (Deng et al., 2023). Our evaluations show significant improvements in task accuracy and adaptability compared to previous methodologies.

## 2 RELATED WORK

**Datasets for GUI Understanding.** Several noteworthy benchmarks have been developed as testbeds for GUI agents, each incorporating test data or GUI environments for evaluating the performance of models. Examples include MiniWob (Shi et al., 2017), MiniWob++ (Liu et al., 2018),

Mind2Web (Deng et al., 2023), and VisualWebArena (Koh et al., 2024), which provide environments for web-based navigation tasks. Additionally, OS-World (Xie et al., 2024) introduces a simulated environment for system operations, while AssistGUI (Gao et al., 2024) focuses on PC software navigation tasks.

In addition to these benchmarks, various publicly available datasets have contributed to training models for UI understanding. For instance, Widget Captioning (Li et al., 2020) offers a dataset that includes mobile UI elements across 88 tasks. RICO (Deka et al., 2017; Sunkara et al., 2022) provides approximately 66k mobile screenshots with view hierarchies, alongside around 500k human annotations that identify various icons by their shapes and semantics. Moreover, datasets such as Pix2Struct (Lee et al., 2023) and Pix2Act (Shaw et al., 2023) have been employed to train models with screen parsing capabilities, leveraging in-house data. Recently, SeeClick (Cheng et al., 2024) has gathered a set of pretraining data to develop an MLLM-based GUI agent.

Nevertheless, these existing datasets are still insufficient in terms of diversity and scale to support the development of a robust GUI grounding model (Lu et al., 2024b). To address this limitation, we have designed an automated data collection engine to assemble a dataset for GUI element grounding, characterized by its high diversity and dense annotations.

**Multimodal LLM for GUI Understanding.** Multimodal Large Language Models (MLLMs) (Liu et al., 2024; Zhu et al., 2023; OpenAI, 2023b; Abdin et al., 2024; Bavishi et al., 2023) have recently made significant advancements, integrating visual perception and language generation to perform tasks such as image captioning (Li et al., 2023; Wang et al., 2020), visual question answering (Hudson & Manning, 2019), and particularly in OCR-free image text comprehension (Luo et al., 2024). The Qwen-VL series (Bai et al., 2023b;a) adds grounding capabilities, allowing models to localize image regions based on language input. LISA (Lai et al., 2024) adds segmentation capabilities to MLLMs by introducing an additional segmentation modules to MLLMs.

These newly added capabilities enable models to better observe the details of images, which supports tasks in GUI environments. As a result, some models (Hong et al., 2023; You et al., 2024; Shaw et al., 2024; Cheng et al., 2024) have been proposed to generate actions in these environments. GUI-based tasks, often requiring the understanding of high-resolution images, are addressed by models like (Cheng et al., 2024; Hong et al., 2023) , enabling interaction with GUIs through visual grounding, thus improving accessibility and widget localization.

While existing models exhibit strong object grounding capabilities, accurately localizing GUI elements in screenshots remains challenging due to their typically small size and similar visual features, such as icons. To overcome this, we designed a specialized decoder, inspired by LISA (Lai et al., 2024), dedicated to grounding GUI elements. Unlike previous approaches that rely solely on pure Transformer architectures to predict bounding boxes for GUI elements (Cheng et al., 2024; Hong et al., 2023; Bavishi et al., 2023), our method addresses the unique challenges posed by GUI screenshots.

## 3 DATA COLLECTION ENGINE

A robust GUI grounding model means strong performance across diverse GUI applications. To achieve this, it is essential to collect a large, high-quality, and varied dataset. The most effective way to gather such a comprehensive dataset is through web screenshots, given their vast quantity and diversity in layouts and elements, which can be generalized to various application scenarios.

Therefore, we propose a data collection engine that automates the browsing of websites and collects high-quality annotations of GUI elements. The collected data includes *screenshots of graphical user interfaces*, *the corresponding GUI elements with basic details (including names and bounding boxes)*, and *the hierarchical information of these elements*. The data we aim to collect through this engine is characterized by:

- **Dense Annotations**: Screens should have comprehensive annotations of elements, including but not limited to types, displayed content, and positional information.
- **Rich Elements**: This includes actionable elements, such as various buttons or controls, as well as static elements like embedded images and static text.
- **Diverse Domains**: The data should cover a broad range of styles and layouts, spanning multiple domains, to ensure versatility and the ability to generalize to various application scenarios.

| Dataset | Task | Platform | Vis. Input | # Inst. | # Anno. | Anno. type |
|---------|------|----------|-----------|---------|---------|------------|
| MiniWob (Shi et al., 2017) | Navigation | Web | Image | 2.8K | 9.8K | traj., pos. |
| Mind2Web (Deng et al., 2024) | Navigation | Web | (multi.) Image | 2.3K | 17K | traj., pos. |
| AITW (Rawles et al., 2023) | Navigation | Mobile | (multi.) Image | 4.6K | 23.6K | traj., pos. |
| AITZ (Zhang et al., 2024) | Navigation | Mobile | (multi.) Image | 2.5K | 18.6K | traj., pos. |
| Odyssey (Lu et al., 2024a) | Navigation | Mobile | (multi.) Image | 7.7K | 119K | traj., pos. |
| GUIWord (Chen et al., 2024a) | Understanding | Diverse | Video | 12.4K | 98K | cap., QA |
| ScreenSpot (Cheng et al., 2024) | Grounding | Web, Desk, Mobile | Image | 1.3K | 1.3K | type, cont., pos. |
| GUIAct (Chen et al., 2024b) | Navigation | Web, Mobile | (multi.) Image | 79K | 191K | type, cont., pos. |
| Screen2words (Wang et al., 2021) | Understanding | Mobile | Image | 79K | 79K | cap.,type,cont.,class,pos,hier. |
| GUIChat (Chen et al., 2024b) | Understanding | Web, Mobile | Image | 50K | 50K | chat, pos. |
| RICO (Deka et al., 2017) | Grounding | Mobile | Image | 243K | 763K | type, cont., pos. |
| GUIEnv (Chen et al., 2024b) | Grounding | Web, Mobile | Image | 70K | 589K | type, cont., pos. |
| AMEX (Chai et al., 2024) | Grounding | Mobile | Image | 97K | 885K | type, cont., pos. |
| SeeClick (Cheng et al., 2024) | Grounding | Web | Image | 0.3M | 3.0M | type, cont., pos. |
| Ours | Grounding | Web | Image | 0.5M | 35M | type,cont.,class,pos,hier. |

Table 1: **Statistics for GUI related datasets.** The upper upper half datasets of smaller size are used for downstream evaluation. The lower half datasets of larger size can be used for pretraining. Understanding type includes chat and/or captioning data. "traj.": trajectory, "pos.": position, "cap": captioning, "QA": quesntion answer, "cont.": displayed content, "class": class type defined by system, "hier": hierarchy info. Among these datasets, our collected data provides the most screenshots and element annotations.

To achieve these goals, our data engine is structured around the following stages:

**Topic Selection.** We follow previous work (Cheng et al., 2024; Hong et al., 2023), sourcing URLs from the Common Crawl. However, many web pages in Common Crawl only contain plain texts with simple layouts, such as pages consisting of only a few lines of text, a few links, or simple images, thus offering few interactive elements. This simplicity creates a significant gap when compared to the complex GUI structures in common GUI tasks. To collect a large number of interactive elements, we selected 11 topics and identified 118 domains related to those topics. These domains were then used to filter URLs from the Common Crawl dataset. The selection of topics and website domains was guided by GPT-4 (Achiam et al., 2023), which proposed a diverse set of topics relevant to real-world applications. This domain list then extracts relevant web page instances from the original Common Crawl dataset. In the Appendix, we display the selected topics and their associated domains.

**GUI Data Collection.** Given the URLs for the selected websites, we built a system that automatically loads these web pages in a web browser, such as Chrome. Since web pages are often lengthy, our system simulates user interaction by pressing the "Page Down" key to scroll through the entire content until the bottom of the page is reached. During this process, we capture screenshots of the web pages along with corresponding element data. The element data, which includes attributes, class names, display contents, and bounding boxes, is obtained using UI Automation (UIA). This approach allows us to comprehensively capture both the visual and structural aspects of the user interface, ensuring that we accurately document and analyze the interactive elements present on each webpage.

**Data Cleaning.** The goal of the data cleaning stage is to ensure the quality and usability of the collected data. Following the gathering of screenshots and element data from various websites, we implement several critical cleaning steps to filter out unusable or irrelevant data, resulting in a high-quality dataset. First, we eliminate websites that are either empty or inaccessible as many URLs may lead to pages with no meaningful content, broken links, or require authentication that cannot be handled automatically. This step is crucial for maintaining the reliability of our dataset. Next, we clean the element data by removing entries with erroneous or missing element names. We also address cases where the element hierarchy is ambiguous or misrepresented by restructuring it to accurately reflect the parent-child relationships between GUI elements. By performing these steps, we refine the dataset to ensure it contains accurate, meaningful, and well-structured data, which is essential for further processing and the effective training of robust GUI grounding models.

**Augmenting GUI Grounding Queries.** To ensure the model's generalization capabilities, it must adapt to various query types and handle a range of tasks such as element identification, grounding, and referring expression. However, the metadata collected primarily provides basic information like class names and display contents, which are insufficient for generating diverse queries.

We address this by generating instructions through a set of defined templates that guide the model to focus on specific aspects of the GUI data. For example, a button labeled "go back" could be

augmented to a query like "Where is the button to go back to the previous page?" These templates, created using GPT-4 after categorizing the collected elements, enable us to generate a diverse set of queries that encompass a wide range of interactions and contexts. Some examples of constructed templates are shown in Appendix. This augmentation process ensures that the model can effectively manage various tasks, improving its ability to generalize across different application scenarios and enhancing its practical usability and robustness.

**Data Overview.** In total, we collected 0.5M screenshots with 35M UI element annotations. Our collected data covers a broad range of topics and features rich elements with dense annotations. We provide a comparison of our collected dataset with previous GUI data in Table. 1.

## 4 MODEL ARCHITECTURE

Within our scope, the GUI grounding model should exhibit generalized capabilities and be easily adaptable for seamless integration with current MLLMs, requiring minimal modifications and ensuring minimal performance loss.

We propose **A**ssist**GUIG**rounder (AGG), a GUI grounding model designed with these objectives in mind. For the GUI grounding model itself, it takes a GUI image and a textual query about the target element as inputs and outputs the position of the target element corresponding to the query. In terms of integrating the GUI grounding model with an MLLM, it should excel in vision-based GUI agent tasks. The inputs include a GUI image representing the current status, a text description of the user goal, and optionally, a history of past actions. The MLLM, acting as a "planner", generates the representation of the action and target element for the next step. Meanwhile, the GUI grounding model extracts visual features from the input image and combines them with the decision information from the MLLM, thus outputting the position of the target element when required.

In the following sections, we will outline the model architecture of the GUI grounding model and discuss how it can be effectively integrated with MLLMs.

### 4.1 GUI GROUNDING MODEL

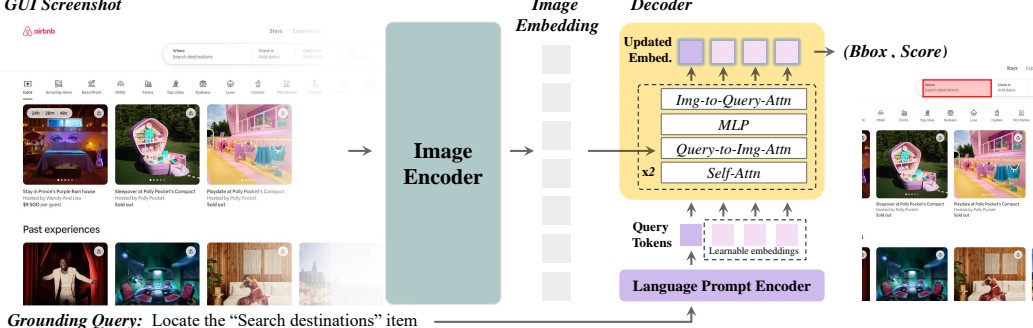

Figure 2: **Overview of GUI Grounding model.** The model consists of three main components: an image encoder, a language prompt encoder, and a lightweight decoder. The image encoder extracts visual features from the input GUI image, while the language prompt encoder transforms the textual query into query embeddings. The lightweight decoder then combines these visual features and query embeddings to predict the position of the target GUI element.

Our GUI grounding model AGG comprises an image encoder $\mathbf{I}$, a language prompt encoder $\mathbf{L}$, and a lightweight decoder $\mathbf{D}$, as illustrated in Figure 2. As an overview, the image encoder extracts the visual features of a GUI image $X_{\text{img}}$, while the language prompt encoder learns to represent the textual query $X_{\text{query}}$. The lightweight decoder then fuses these visual features and the textual query representation, learning to predict the position of the target GUI element, denoted as $\hat{b}$. This can be formulated as: $\hat{b} = \mathbf{D}\left(\mathbf{I}(X_{\text{img}}), \mathbf{L}(X_{\text{query}})\right)$. Below is a detailed description of these components:

**Image Encoder.** We leverage the scalable pretraining benefits by employing the SAM (Kirillov et al., 2023b) pretrained Vision Transformer (ViT) (Dosovitskiy et al., 2021). This choice is made due to its demonstrated effectiveness in promptable point-to-mask segmentation tasks, showcasing its strong

visual perception capabilities. For handling high-resolution GUI inputs, we utilize windowed attention along with four equally spaced global attention blocks, following the methodology in (Kirillov et al., 2023b; Li et al., 2022).

**Language Prompt Encoder.** The language encoder is applied to maps text queries into query embeddings. Specifically, we employ a language transformer model (Devlin et al., 2019) as the prompt encoder. We prepend <CLS> to each text query and use its output hidden states from the final transformer layer as the query embedding.

**Lightweight decoder.** The decoder's role is to process the embeddings from both the image encoder and the language prompt encoder, mapping them to the output bounding boxes which indicate the precise locations of GUI elements. Our decoder architecture is inspired by the design principles outlined in SAM (Kirillov et al., 2023b). Before inputting the embeddings into the decoder, we introduce a set of learnable embeddings and concatenate them with the query embeddings from the prompt encoder. These combined embeddings are collectively referred to as "query tokens".

The decoder comprises two layers, each performing a series of operations: (1) Bi-directional self-attention on the query tokens, allowing the model to convey information within the tokens; (2) Cross-attention from query tokens (as *query*) to the image embedding; (3) An MLP updates the query tokens; (4) Cross-attention from the image embedding (as *query*) to the query tokens, which updates the image embeddings with contextual information derived from the queries. The next decoder layer then takes the updated query tokens and the updated image embeddings from the previous layer. Finally, the output of the query tokens from the last decoder layer is used for grounding prediction.

**Training for Grounding.** For the GUI image input, we follow (Kirillov et al., 2023b) to rescale the input image to $1024 \times 1024$, by adjusting the long side while maintaining the aspect ratio and padding the short side. For the textual query input, to enhance the generalizability of the grounding model, we augment the text queries by sampling templates from a predefined set for different elements. For instance, the metadata from the data collection engine might only provide structured information such as `{"class_name":"button","content":"go back","level-info":["navigation bar","go back"],...}` for a button. Instead of merely concatenating this information into "button: go back", we generate more diverse queries like "Seek the go back button and click", or "Help me find the go back button in the navigation bar".

Following object detection methods (Carion et al., 2020), we add prediction heads to the decoder outputs for bounding box prediction. Each output from the learnable embeddings predicts both a score and bounding box coordinates. The predicted score is optimized to match the IoU of the predicted and ground-truth boxes using mean-square-error loss $\mathcal{L}_{\mathrm{mse}}$. For bounding box regression, we use a linear combination of $l_1$ loss $\mathcal{L}_{l_1}$ and the generalized IoU loss $\mathcal{L}_{\mathrm{GIoU}}$ (Rezatofighi et al., 2019). The overall training loss for GUI grounding is given by: $\mathcal{L} = \lambda_{\mathrm{mse}}\mathcal{L}_{\mathrm{mse}} + \lambda_{l_1}\mathcal{L}_{l_1} + \lambda_{\mathrm{GIoU}}\mathcal{L}_{\mathrm{GIoU}}$, where $\lambda_{\mathrm{mse}}$, $\lambda_{l_1}$, and $\lambda_{\mathrm{GIoU}}$ are hyper-parameters. Unlike (Kirillov et al., 2023b), which backpropagates only from the lowest loss, we found that backpropagating from all predictions improves performance in GUI grounding. During inference, bounding boxes are ranked by score, and the top-1 box is selected as the final prediction.

## 4.2 Integrating Grounding Model with MLLM as Agent

For vision-based agents, completing a complex GUI task typically involves several steps, each requiring the model to: (1) Perceive and understand: gather and interpret information from GUI images, user goals, and action history to understand the current state. (2) Reason and plan: analyze the current state to decide on the next action. (3) Execute: perform the desired action on the target GUI element. In this work, we propose the simple yet effective embeddings-as-position paradigm to integrate the GUI grounding model's capabilities into an MLLM, thereby creating a vision-based agent tailored for these tasks. We detail the design below.

**Architecture for the integrated agent.** The proposed agent consists of an MLLM M and a well-trained GUI grounding model, as is shown in Figure. 3. Inspired by (Lai et al., 2024), this approach involves several key steps. Initially, we add a new token <|POS|> to the original MLLM vocabulary, indicating a request for a GUI grounding output. Given the textual input $X_{\mathrm{usr}}$, which includes the user goal and optionally, the action history, along with the visual input of the GUI image $X_{\mathrm{img}}$, these inputs are fed into the MLLM, which generates a text-form response $\hat{o}_{\mathrm{usr}}$. This process is formulated

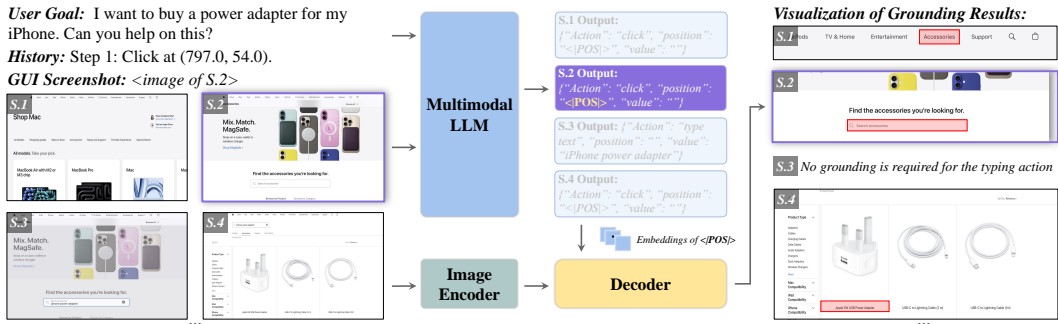

Figure 3: **Integrate AGG with MLLM as a GUI agent.** The proposed agent integrates an MLLM with a trained GUI grounding model. MLLM processes textual inputs (user goals, action history) and visual inputs (GUI screenshot) to generate a response. A special token triggers GUI grounding. The image encoder extracts visual features, then the lightweight decoder predicts the target GUI element's position. The figure illustrates the process of task completion and highlights the step 2.

as $\hat{o}_{\mathrm{usr}} = \mathbf{M}(X_{\mathrm{img}}, X_{\mathrm{usr}})$. When the MLLM aims to generate a bounding box, the text-form output includes a `<|POS|>` token. We then extract the last layer embedding corresponding to the `<|POS|>` token and apply a linear projection layer to it. We denote this output as $h_{\mathrm{pos}}$. Meanwhile, the image encoder from the grounding model extracts visual features from the input GUI image. Finally, these visual features and the projected embedding are fed into the lightweight decoder of the grounding model to predict the position $P$ of the desired GUI element, formulated as $P = \mathbf{D}\left(h_{\mathrm{pos}}, \mathbf{I}(X_{\mathrm{img}})\right)$.

**Training for agent tasks.** The model is trained using the text token prediction loss $\mathcal{L}_{\mathrm{text}}$ and the grounding loss $\mathcal{L}_{\mathrm{grd}}$. The overall objective for training the vision-based agent is: $\mathcal{L} = \lambda_{\mathrm{text}}\mathcal{L}_{\mathrm{text}} + \lambda_{\mathrm{grd}}\mathcal{L}_{\mathrm{grd}}$, where $\lambda_{\mathrm{text}}$ and $\mathcal{L}_{\mathrm{grd}}$ are hype-parameters. Especially, when the form of the target position is a bounding box, $\mathcal{L}_{\mathrm{grd}}$ is the same as that used in GUI grounding training. When the form of the target position is point, we slightly modify the prediction head upon the decoder output to predict the normalized distance to the ground truth point as a score indicator, and predict the 2D point coordinates as the position output.

## 5 EXPERIMENT

### 5.1 EXPERIMENTAL SETUP

**Training data for GUI Grounding.** We train our GUI grounding model on two datasets: web UI data collected by our engine and mobile UI data reorganized from public sources. For web UI, we applied a data smoothing strategy (Hudson & Manning, 2019), resulting in 0.36M images with 22M element annotations. For mobile UI, following (Cheng et al., 2024), we used 27K screenshots and 275K query-bbox annotations from the WidgetCaption (Li et al., 2020) and RICO (Deka et al., 2017).

**Training for Agent-based Tasks.** We integrate AGG with an MLLM using the embeddings-as-position approach. We verify its efficacy on agent-based tasks, including MiniWob (Shi et al., 2017), AITW (Rawles et al., 2023), and Mind2Web (Deng et al., 2024). Following (Cheng et al., 2024), we pretrain Qwen-VL (Bai et al., 2023b) using our dataset combined with a mixed public dataset to build an MLLM for GUI tasks. We then combine it with our trained grounding model to finetune on each agent-based task. Instructions and previous action memory are provided during both training and inference, as described in (Cheng et al., 2024).

### 5.2 EXPERIMENTS FOR GROUNDING AND AGENT TASKS

***ScreenSpot.*** ScreenSpot is a benchmark for GUI grounding. It comprises over 600 interface screenshots and more than 1200 annotations for query-element pairs. It spans mobile platforms (iOS, Android), desktop platforms (macOS, Windows), and web platforms.

For this benchmark, we compare our GUI grounding model with generalist MLLMs capable of recognition and reasoning (Chen et al., 2023; Bai et al., 2023b; Yan et al., 2023; Abdin et al., 2024), as well as recent MLLMs that have incorporated GUI-related tasks during pretraining (Bavishi et al., 2023; Hong et al., 2023; Cheng et al., 2024).

*Results.* The results for grounding, shown in Table. 2, highlight that while generalist MLLMs possess extensive knowledge about natural images, their GUI grounding performance on ScreenSpot is subpar due to the significant differences between GUIs and natural images. Even GPT-4V struggles to accurately locate screen elements. GUI-specific MLLMs demonstrate better performance. In particular, Our AGG achieves the highest performance across different platforms and for both text and icon elements, even with fewer model parameters. This underscores the advantages of our collected data for GUI grounding training, allowing AGG to effectively handle diverse queries and rich elements encountered during training. With rich elements and diverse query seen during training, AGG possess the capability for GUI grounding, even for the very less scenarios like iOS. Compared to text element grounding, the model's performance is less strong on icon-based tasks, indicating that grounding non-text elements in GUIs remains challenging.

| Method | GUI.S | Model size | Mobile | | Desktop | | Web | | Avg. |
| --- | --- | --- | --- | --- | --- | --- | --- | --- | --- |
| | | | Text | Icon | Text | Icon | Text | Icon | |
| MiniGPTv2 (Chen et al., 2023) | ✗ | 7B | 8.4 | 6.6 | 6.2 | 2.9 | 6.5 | 3.4 | 5.7 |
| Qwen-VL (Bai et al., 2023b) | ✗ | 9.6B | 9.5 | 4.8 | 5.7 | 5.0 | 3.5 | 2.4 | 5.2 |
| Phi-3.5-V (Abdin et al., 2024) | ✗ | 4.2B | 1.7 | 1.7 | 4.1 | 0.7 | 4.7 | 2.0 | 2.5 |
| GPT-4V OpenAI (2023a) | ✗ | - | 22.6 | 24.5 | 20.2 | 11.8 | 9.2 | 8.8 | 16.2 |
| Fuyu (Bavishi et al., 2023) | ✓ | 8B | 41.0 | 1.3 | 33.0 | 3.6 | 33.9 | 4.4 | 19.5 |
| CogAgent (Hong et al., 2023) | ✓ | 18B | 67.0 | 24.0 | 74.2 | 20.0 | 70.4 | 28.6 | 47.4 |
| SeeClick (Cheng et al., 2024) | ✓ | 9.6B | 78.0 | 52.0 | 72.2 | 30.0 | 55.7 | 32.5 | 53.4 |
| AGG (Ours) | ✓ | 0.4B | **86.1** | **62.8** | **81.8** | **46.2** | **74.2** | **48.4** | **66.6** |

Table 2: **Performance of various models on ScreenSpot.** "GUI.S" indicates "GUI Specific", denoting whether the model was trained using domain-specific data.

*MiniWob.* MiniWob (Shi et al., 2017) features 2000 open-ended tasks from 137 real web environments, each with high-level instructions and action trajectories. These tasks require the agent to navigate a simplified web environment and execute human-given instructions.

Following (Cheng et al., 2024), we conduct 2.8K episode rollouts for training. We compared our method with models using image-only inputs, text-only inputs, and a combination of both. Note that with text input from the environment, the model selects the HTML element from candidates as the action target. We follow (Cheng et al., 2024) and report performance on two groups of data splits. We compute the success rate over 50 random seeds for each task and then calculate the mean success rate across all tasks as the final score.

| Method | Modality | Dataset | Score |
| --- | --- | --- | --- |
| CC-Net(SL) (Humphreys et al., 2022) | DOM+Image | 2.4M | 36.5 |
| WebN-T5 (Gur et al., 2022) | HTML | 12K | 55.2 |
| WebGUM (Furuta et al., 2024) | HTML+Image | 2.8K | 65.5 |
| WebGUM (Furuta et al., 2024) | HTML+Image | 347K | **86.1** |
| SeeClick (Cheng et al., 2024) | Image | 2.8K | 73.6 |
| MLLM ∘ AGG (Ours) | Image | 2.8K | 75.8 |

Table 3: **Results on MiniWob 45-tasks split.**

| Method | Modality | Dataset | Score |
| --- | --- | --- | --- |
| CC-Net(SL) (Humphreys et al., 2022) | Image | 2.4M | 23.4 |
| Pix2Act (Shaw et al., 2024) | Image | 12K | 55.2 |
| Qwen-VL (Bai et al., 2023b) | Image | 2.8K | 48.4 |
| SeeClick (Cheng et al., 2024) | Image | 2.8K | 67.0 |
| MLLM ∘ AGG (Ours) | Image | 2.8K | **69.3** |

Table 4: **Results on MiniWob 35 tasks split.**

*Results.* We present results in Table. 3 and Table. 4. We denote our method as "MLLM ∘ AGG". The results show that our vision-based agent surpasses previous methods. Notably, with 2.8K training episodes, it outperforms WebGUM, which uses both HTML and GUI images as input and exceeds the vision-based Pix2Act, which used less than 0.3% as much training data. Furthermore, our method outperforms MLLMs like Qwen-VL and SeeClick, demonstrating the efficacy of our framework.

*AITW.* Android In The Wild (AITW) (Rawles et al., 2023) dataset is a mobile navigation benchmark including 30K instructions and 715K corresponding operation trajectories. Following the approach detailed in (Cheng et al., 2024), we apply the same train/test split based on instructions, retaining a single trajectory per instruction and ensuring no overlap between the training and test sets.

We compare our method to API-based LLMs and previous MLLM-based methods. We use the screen-wise action matching score as the main metric, as outlined in (Rawles et al., 2023), and follow (Cheng et al., 2024) to compute click accuracy to evaluate grounding capabilities.

*Results.* As illustrated in Table. 5, our method achieved the best performance across all tasks, surpassing both the API-based LLMs and the previous GUI data pretrained MLLMs. Notably, our method exhibits higher click accuracy than other methods, demonstrating stronger GUI grounding capabilities.

| Method | Modality | General | Install | GoogleApps | Single | WebShopping | Overall | ClickAcc |
|---|---|---|---|---|---|---|---|---|
| ChatGPT-CoT (Zhang & Zhang, 2024) | Text | 5.9 | 4.4 | 10.5 | 9.4 | 8.4 | 7.7 | - |
| PaLM2-CoT (Rawles et al., 2023) | Text | - | - | - | - | - | 39.6 | - |
| GPT-4V (Yan et al., 2023) | Image | 41.7 | 42.6 | 49.8 | **72.8** | 45.7 | 50.5 | - |
| Qwen-VL Bai et al. (2023b) | Image | 49.5 | 59.9 | 46.9 | 64.7 | 50.7 | 54.3 | 57.3 |
| SeeClick Cheng et al. (2024) | Image | 54.0 | 66.4 | 54.9 | 63.5 | 57.6 | 59.3 | 66.4 |
| MLLM ∘ AGG (Ours) | Image | **56.4** | **67.8** | **56.6** | 67.2 | **63.1** | **62.2** | **68.1** |

Table 5: **Performance of different models on AITW.** The upper half corresponds to the performance of API-based LLMs.

| Method | PT | Cross-Task | | | Cross-Website | | | Cross-Domain | | |
|---|---|---|---|---|---|---|---|---|---|---|
| | | Ele.Acc | Op.F1 | Step.SR | Ele.Acc | Op.F1 | Step.SR | Ele.Acc | Op.F1 | Step.SR |
| MindAct (Gen) (Deng et al., 2024) | – | 20.2 | 52.0 | 17.5 | 13.9 | 44.7 | 11.0 | 14.2 | 44.7 | 11.9 |
| MindAct (Deng et al., 2024) | – | 55.1 | 75.7 | 52.0 | 42.0 | 65.2 | 38.9 | 42.1 | 66.5 | 39.6 |
| GPT-3.5-Turbo | – | 20.3 | 56.6 | 17.4 | 19.3 | 48.8 | 16.2 | 21.6 | 52.8 | 18.6 |
| GPT-4 (OpenAI, 2023a) | – | 41.6 | 60.6 | 36.2 | 35.8 | 51.1 | 30.1 | 37.1 | 46.5 | 26.4 |
| SoM* (Yang et al., 2023a) | ✗ | 29.4 | 67.5 | 22.6 | 19.4 | 54.2 | 20.7 | 23.8 | 60.3 | 24.2 |
| SeeAct(Anno)* (Zheng et al., 2024) | ✗ | - | - | 20.3 | - | - | 13.9 | - | - | 23.7 |
| SeeAct(Choice)[†] (Zheng et al., 2024) | ✗ | 46.4 | 73.4 | 40.2 | 38.0 | 67.8 | 32.4 | 42.4 | 69.3 | 36.8 |
| Qwen-VL (Bai et al., 2023a) | ✗ | 15.9 | 86.7 | 13.3 | 13.2 | 83.5 | 9.2 | 14.1 | 84.3 | 12.0 |
| SeeClick (Cheng et al., 2024) | ✓ | 28.3 | 87.0 | 25.5 | 21.4 | 80.6 | 16.4 | 23.2 | 84.8 | 20.8 |
| MLLM ∘ AGG (Ours) | ✓ | **41.2** | **87.4** | **36.8** | **39.2** | **84.8** | **33.8** | **38.9** | **85.2** | **34.2** |

Table 6: **Performance of different models on Mind2Web.** "Ele.Acc" denotes element accuracy, "Op.F1" denotes operation F1 score and "Step.SR" denotes step success rate. "PT" indicates if the MLLM is pretrained on GUI data. *SoM and SeeAct(Anno) are tested on partial test set. [†]SeeAct(Choice) uses an image and text candidates from a ranking LM as inputs.

**Mind2Web.** Mind2Web (Deng et al., 2024) is a benchmark that evaluates an agent's capabilities in web navigation tasks. It encompasses over 2000 open-ended tasks sourced from 137 real website URLs, each accompanied by high-level instructions and corresponding human action trajectories.

In this work, we follow (Cheng et al., 2024) to use parsed screenshots and target element bounding boxes from the raw HTML data of Mind2Web. We compare our approach with both text-based web agents (Deng et al., 2024; OpenAI, 2023a) and vision-based agents (Bai et al., 2023b; Cheng et al., 2024; Yang et al., 2023a; Zheng et al., 2024). Mind2Act employs a two-step process: a small language model first generates candidate elements from raw HTML, followed by a large language model that selects the target via multiple-choice QA. Mind2Act (gen) directly generates the target element. GPT-3.5 and GPT-4 use the same multiple-choice QA formulation and include three demonstrations for in-context learning. SoM overlays label indices for element proposals on the input image and prompts GPT-4V to select the matching label. SeeAct (Anno) overlays labels at ground truth positions of candidate elements predicted by a ranking language model. For SeeAct (Choice), GPT-4V selects an answer from candidate text choices predicted by a ranking language model. Qwen-VL, SeeClick and our model are fine-tuned using trajectory annotations.

Following the evaluation metrics from (Cheng et al., 2024; Deng et al., 2024), we measure element accuracy, operation F1, and step success rate (StepSR). For vision-based models, a prediction is considered correct if the predicted point (or the center of the predicted bounding box) falls within the target element's ground-truth bounding box.

*Results.* In Table. 6, the results on Mind2Web demonstrate that pretraining on GUI grounding data significantly enhances agent performance. Our model outperforms SoM and SeeAct (Anno), both of which struggle with accurately proposing element regions or associating GPT-4V's action output with the correct label (Zheng et al., 2024). Our method also surpasses the previous state-of-the-art vision-based agent, SeeClick, by a large margin in terms of element accuracy, underscoring the robust grounding capabilities of our model. However, our model still falls short of the top-performing text-based GUI agents, and the text-candidate-elements involved SeeAct (Choice), as grounding elements in images is inherently more complex than selecting an element from multiple choices. This highlights the challenges associated with GUI element grounding and suggests that there remains substantial room for improvement in deploying vision-based GUI agents for real-world applications.

## 5.3 Ablation Study

In this section, we conduct ablation studies to analyze the effectiveness of various design elements in the grounding model and vision-based GUI agent.

| Model Var | avg. ele.acc |
|---|---|
| Qwen-VL | 14.4 |
| Qwen-VL ∘ AGG | 27.4 |
| Qwen-VL(SC data) | 24.3 |
| Qwen-VL(SC data) ∘ AGG | 35.2 |
| Qwen-VL(our data) | 30.7 |
| Qwen-VL(our data) ∘ AGG | 39.8 |

Table 7: Effective factors for agent tasks: different data to pretrain the MLLM and whether incorporate AGG. The default setting is marked with a color box .

| Training config | avg. |
|---|---|
| #1. Random init, Full | 41.1 |
| #2. SAM init, Lora | 46.8 |
| #3. prop min | 47.3 |
| #4. prop max | 48.6 |
| #5. w/o aug | 44.5 |
| #6. *default* | 49.2 |

Table 8: Training configuration for AGG training. Default setting is marked with color box .

| # prompt emb. | avg. |
|---|---|
| 1 | 46.2 |
| 2 | 48.4 |
| 3 | 49.2 |
| 4 | 49.0 |
| 5 | 49.8 |

Table 9: Number of prompt embedding for decoder. Default setting is marked with color box .

**Effective factors for improving agent tasks.** We assess the impact of our collected data and AGG. Specifically, we evaluate the Qwen-VL (Bai et al., 2023b) MLLM without GUI data pretraining, trained with SeeClick's pretraining data, and trained with our pretraining data. We then integrate AGG to these model variants. The average element accuracy on Mind2Web (Deng et al., 2024) is reported. Results in Table. 7 indicate that using our collected data during pretraining results in an MLLM with superior GUI grounding capabilities. Additionally, integrating AGG leads to further improvements, demonstrating the effectiveness of our proposed paradigm.

Next, we investigate the key factors influencing the effectiveness of GUI grounding training. We use the ViT-SAM-Base configuration for the grounding model and a fixed seed to randomly sample 25% of the collected GUI images and their corresponding annotations for training. We report the average grounding accuracy on ScreenSpot.

**Training Configurations for the Image Encoder**. We examine the impact of using SAM model weights for initialization and different tuning strategies for the image encoder during training. The results in Table. 8 (#1,2,6) demonstrate that initializing the image encoder with SAM model weights and fully tuning it during training yields the best performance.

**Grounding Loss Propagation During Training**. We examine the impact of loss propagation configurations for GUI grounding training. The results shown in Table. 8 (#3,4,6) indicate that propagating the loss from all prompt embeddings yields the best performance for GUI grounding.

**Effectiveness of augmenting prompt queries**. We investigate the effect of augmenting the query from metadata during GUI grounding training. Results in Table. 8 (#5,6) show that augmenting query significantly enhance the GUI grounding capability, which is important to build a generalized vision-based GUI agent.

**Number of Learnable Prompt Embeddings for Decoder**. We investigate the impact of the number of learnable prompt embeddings used in the decoder. Table. 9 shows that increasing the number of prompts beyond 3 yields no significant improvement in grounding performance.

## 6 Conclusion

In this work, we investigated the critical challenge of precise GUI element grounding, which is essential for accurate interaction and generalization in GUI agents. We developed a simple yet effective GUI grounding framework that includes an automated data collection engine to gather extensive GUI screenshots and annotations, ensuring broad generalization. We introduced a lightweight GUI grounding module designed for efficient UI element localization by pre-training on the collected data. By integrating this module with MLLMs, we demonstrated superior performance in task accuracy and adaptability across multiple benchmarks, including ScreenSpot, MiniWob, AITW, and Mind2Web. Our results highlight the effectiveness of our approach in enhancing the capabilities of GUI agents for complex tasks on digital platforms.

ETHICS STATEMENT

This research involves the use of a training dataset collected from publicly available web data for the purpose of training GUI-based AI models. The dataset used in this paper was sourced from Common Crawl, which provides data under specific term of use. According to these terms, users are permitted to copy, modify, distribute, and use the data. Our use of the data is transformative in nature, focusing on creating novel AI models for GUI understanding.

We have ensured that the data utilization complies with all applicable laws, ordinances, rules, and regulations. We have processed and filtered the collected data to ensure it contains no sensitive or personally identifiable information. By adhering to Common Crawl's terms of use, we confirm that our data usage is legally permissible and meets the ethical standards set forth by both Common Crawl and the research community.

Additionally, we acknowledge that we are solely responsible for any legal violations concerning the collected data and accept all associated risks. We have implemented measures to ensure that the data collection process respects privacy, copyright, and other legal constraints.

Furthermore, any distribution of the data will adhere to relevant regulations and guidelines, ensuring the protection of privacy and security, and compliance with AI ethics standards.

ACKNOWLEDGEMENTS

This research is supported by the National Research Foundation, Singapore under its AI Singapore Programme (AISG Award No: AISG3-RP-2022-030). The authors would like to thank Kevin Qinghong Lin for his valuable discussions and insightful feedback on this project, and Peiran Li, Xiangwu Guo, and Qinchen Wu for their assistance in data collection.

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

APPENDIX

## A MORE INFORMATION FOR THE COLLECTED DATA

In this section, we provide additional details about our collected data.

**Topics and Domains.** We list the topics and some sampled domains in our collected data, as is shown in Table. 10.

| Topic | Related Domains |
|---|---|
| News and Journalism | `www.bbc.com, www.cnn.com, ...` |
| Technology and Science | `www.wired.com, www.sciencedaily.com, ...` |
| Sports | `www.espn.com, www.cbssports.com, ...` |
| Entertainment | `www.imdb.com, www.netflix.com, ...` |
| Lifestyle and Wellness | `www.wellandgood.com, www.goop.com, ...` |
| Shopping and E-commerce | `www.amazon.com, www.ebay.com, ...` |
| Travel | `www.airbnb.com, www.booking.com, ...` |
| Food and Cooking | `tasty.co, www.foodnetwork.com, ...` |
| DIY and Home Improvement | `www.diynetwork.com, www.houzz.com, ...` |
| Educational and Learning | `www.coursera.org, www.ted.com, ...` |
| Others | `www.github.com, www.capcut.com, ...` |

Table 10: Examples topics and some corresponding domains in our collected data.

**Data Example Visualization.** In Figure. 4, we present several samples from our collected dataset. These visualizations include bounding boxes highlighting the corresponding elements.

**Examples of Augmented Query Templates.** In Table. 11, we showcase a portion of the augmented query templates used to generate queries for GUI grounding training. These templates include both common templates and element-specific templates. During training, the query is generated using either the common templates or the element-specific templates.

**Statistics of Web GUI Data**. Figure. 5 presents the distribution of the number of instances (elements) in our collected Web GUI dataset. It shows that most web screenshots in our dataset contain 20-100 annotated elements, highlighting the "dense annotation" characteristic of our dataset.

Figure. 6 illustrates the distribution of element types in our collected dataset. We combined the same element types from the navigation bar and the main content areas of the websites. The "Static" element, which primarily includes text elements, ranks first in the dataset, likely enhancing grounding performance for text elements. The "Button" element ranks second, which may also contribute to improved grounding performance for this type. Our collected data encompass a broad range of elements, showcasing the "diverse and rich annotation" characteristic of our dataset.

Looking towards future directions, data construction for GUI Agents could focus on building high-quality SFT data across different platforms, applications, and scenarios (Wang et al., 2024a; Lin et al., 2024).

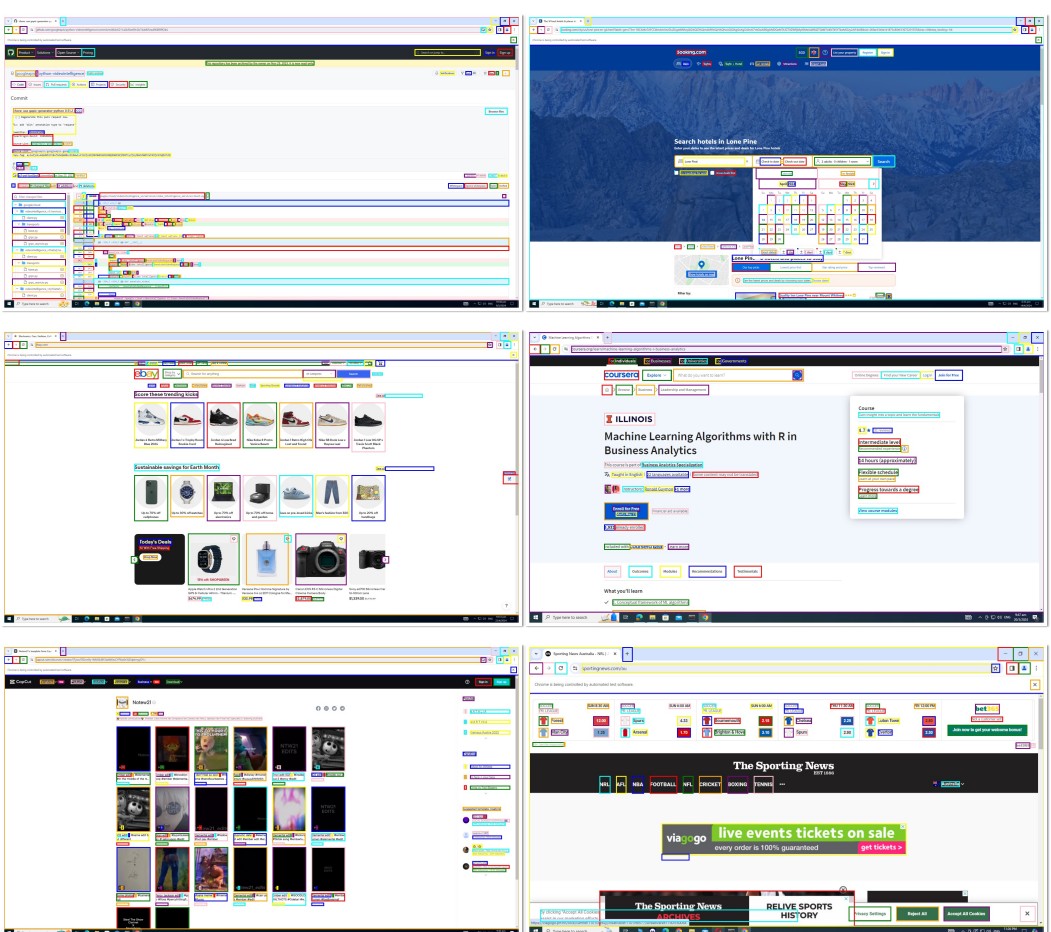

Figure 4: Data examples from our collected data.

**Augmented Query Examples.**

**Examples for common queries**
○ "Look for {content}",
○ "Identify the element {content}",
○ "Locate the {class name} called {content}",
○ "Scroll to the {class name} with {content}",
○ "Highlight the {class name} that contains {content}",
○ "Focus on the {class name} that displays {content}",
○ "Locate the {class name} that is part of {parent info} with the name {content}" ...

**Examples for specific element (button)**
○ "Click the {content} button",
○ "Press the button labelled {content}",
○ "Activate the {content} button",
○ "Find and click the {content} on the page",
○ "Where is the {content} button located?",
○ "Can you locate the button that says {content}?",
○ "Click the {content} button in the {parent info}" ...

**Examples for specific element (DataItem)**
○ "Tick the {content} checkbox",
○ "Check the {content} option",
○ "Mark the checkbox for {content}",
○ "Select the {content} checkbox from the list",
○ "Ensure the {content} checkbox is checked",
○ "Activate the checkbox labeled {content}",
○ "Verify that the {content} checkbox is selected" ...

**Examples for specific element (checkbox)**
○ "Identify the data item labeled {content}",
○ "Locate the {content} data field on the page",
○ "Find the {content} item in the dataset",
○ "Access the information for {content}",
○ "Highlight the data item {content}",
○ "Explore the {content} item for more info" ...

**Examples for specific element (hyperlink)**
○ "Use the {content} link to access the desired page",
○ "Identify the link called {content} and click on it",
○ "Proceed by using the hyperlink {content}",
○ "Check the {content} link for more details",
○ "Open the web page linked by {content}",
○ "Access resources through the {content} hyperlink",
○ "Click the {content} URL to navigate",
○ "Follow the online pointer marked as {content}" ...

**Examples for specific element (static text)**
○ "Highlight the statement {content}",
○ "Observe the static text {content} on this page",
○ "Search for the label {content} in the content",
○ "Inspect the static field titled {content}",
○ "See the displayed static text {content}",
○ "Look for the static description {content}",
○ "Spot the phrase {content} in the view",
○ "Find the static wording {content}" ...

Table 11: Augmented query examples during GUI grounding training. Shown are examples of some of the augmented templates.

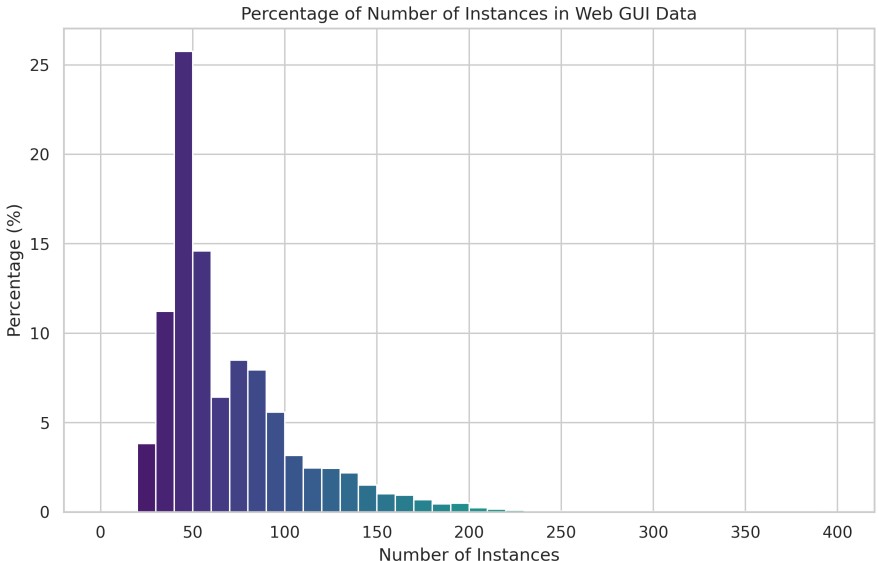

Figure 5: Distribution of "Numer of instances (elements) per GUI image" of our collected Web GUI dataset.

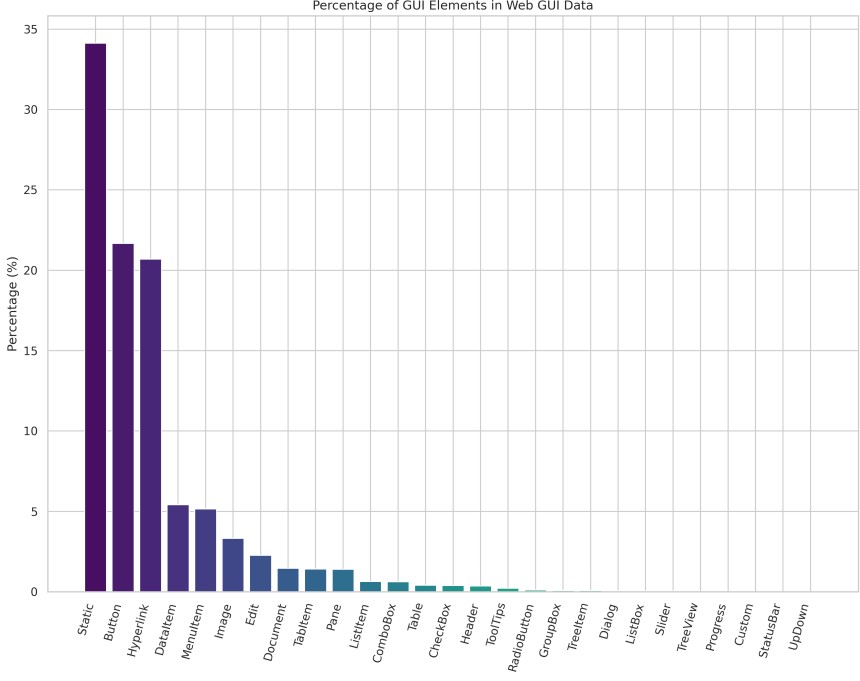

Figure 6: Distribution of "element type" of our collected Web GUI dataset.

## B    ETHICAL CONSIDERATIONS OF THE COLLECTED DATASET

**Discussion on NSFW Content Filtering.** To ensure that our dataset does not contain NSFW content, we perform several steps. Before collection, we carefully screen candidate websites to filter out those that may contain NSFW content. During the data collection process, we also apply manual reviews for verification to ensure the dataset is suitable for academic research and deployment without the risk of including inappropriate material.

**Discussion on Copyright.** Our Web GUI data is sourced from the internet, and to address potential copyright issues, we ensure compliance with copyright laws and regulations by:

- Sourcing Data from Public Domains: We primarily collect data from publicly accessible websites explicitly available for educational and research purposes.

- Respecting Website Policies: We adhere to the terms of service and data use policies of the websites we crawl.

- Anonymizing and Aggregating Data: We anonymize and aggregate the collected data, focusing on general patterns and characteristics to minimize potential copyright infringements.

**Statement of Responsibility and Risk Acknowledgment:** The authors explicitly acknowledge and accept full responsibility for any legal implications arising from the collection and usage of the benchmark datasets in this work. We hereby declare that we understand and accept all associated risks regarding data usage, copyright compliance, and legal responsibilities.

## C    ADDITIONAL EXPERIMENTS

In this section, we present further details, experiments, discussions, and analyses to demonstrate the effectiveness of our model and explore future directions.

**Implementation details.** We configure the image encoder and lightweight decoder of the GUI grounding model using the ViT-SAM-Large (Kirillov et al., 2023b) and the language prompt encoder with a BERT-based (Devlin et al., 2019) by default. Training runs on 32 V100 GPUs with a global batch size of 128 for 150K steps. The AdamW (Loshchilov, 2017) optimizer is used with $\beta_1 = 0.9$, $\beta_2 = 0.98$, and a weight decay of 1e-4. A Cosine Annealing scheduler manages the learning rate, starting with a warm-up over 200 steps. The max learning rate is 1e-3, dropping to 5e-5. Hyperparameters for the training objective are set as $\lambda_{\mathrm{mse}} = 10$, $\mathcal{L}_{l_1} = 5$, and $\lambda_{\mathrm{GIoU}} = 2$. For vision-based agent tasks, we fine-tune with 2 A100 GPUs, applying LoRA (Hu et al., 2021) tuning (rank 8, alpha 16) for the language model. The decoder parameters are unfrozen, while the image encoder remains frozen. We use AdamW with $\beta_1 = 0.9$, $\beta_2 = 0.98$, and no weight decay. The training objective hyperparameters are $\lambda_{\mathrm{text}} = 1$ and $\mathcal{L}_{\mathrm{grd}} = 1$. For vision-based agent tasks, we fine-tune with 2 A100 GPUs, applying LoRA (Hu et al., 2021) tuning (rank 8, alpha 16) for the language model. The decoder parameters are unfrozen, while the image encoder remains frozen. We use AdamW with $\beta_1 = 0.9$, $\beta_2 = 0.98$, and no weight decay. The training objective hyperparameters are $\lambda_{\mathrm{text}} = 1$ and $\mathcal{L}_{\mathrm{grd}} = 1$.

**Scaling Pretraining Data for GUI Grounding.** We explored the effect of scaling our collected web GUI data on model performance. Using the base-size model (ViT-SAM-Base), we conducted experiments by sampling 25%, 50%, 75%, and 100% of the training data from websites within the same domain. We then evaluated the model's grounding performance on ScreenSpot.

Results in Table. 12 indicate that increasing the amount of training data leads to better grounding performance. However, we observed a slowdown in performance improvement when scaling from 75% to 100% compared to the gains from 25% to 50% and 50% to 75%. Despite this, the data scaling paradigm still shows potential for further improvement. Future efforts can focus on collecting more diverse and high-quality GUI data to enhance performance even further.

**Comparison with Text-involved Method.** Both SeeAct (Choice) (Zheng et al., 2024) and MindAct (Deng et al., 2024) adopt text-form candidates as input and select an answer from them. Although this approach is easier than visually grounding the target element using only an image, these methods are still state-of-the-art for web agent tasks. However, they face challenges with similar or identical elements, which are common on webpages. The model often selects the first text choice that

| Data scale | Mobile | | Desktop | | Web | | **Avg.** |
|---|---|---|---|---|---|---|---|
| | Text | Icon | Text | Icon | Text | Icon | |
| 25% | 72.4 | 50.8 | 59.4 | 27.3 | 57.5 | 28.0 | 49.2 |
| 50% | 78.2 | 53.5 | 68.4 | 31.9 | 62.8 | 33.5 | 54.7 |
| 75% | 79.6 | 54.8 | 74.5 | 35.2 | 66.2 | 36.8 | 57.9 |
| 100% | 80.2 | 56.1 | 75.9 | 39.1 | 68.2 | 41.6 | 60.2 |

Table 12: Results of scaling pretraining data for ScreenSpot grounding.

appears to correspond to its intent. This issue is inevitable since many webpage elements may have identical HTML attributes.

Our model addresses this problem effectively. In Figure. 7, (A) shows a task for "schedule a repair service" and (B) depicts a task for "rent a truck". With identical element names provided as input, it is impossible for an LLM to definitively identify the correct one. As a result, the text-involved methods failed to select the correct button, as shown in the red boxes. In contrast, our model successfully grounds the correct position to click, as indicated by the green dots in Figure. 7. This success is due to the direct integration of AGG, which allows the context from the MLLM to be effectively passed to the grounding module for context-aware grounding, thereby avoiding the information loss seen in SeeAct (Choice) and MindAct.

**Evaluation on Desktop Benchmark.** To test the our model's abilities on software or OS level, we use the Desktop split of the OmniAct benchmark (Kapoor et al., 2024). This benchmark covers 38 desktop applications across different desktop platforms, and requires the model to generate a piece of PyAutoGUI script at each step to complete the task. We compare our method with SeeClick (Cheng et al., 2024) and SoM (Yang et al., 2023a). We finetuned models on the OmniAct training set and followed (Kapoor et al., 2024) to measure the Action Score (AS) and include the grounding successful rate for steps requiring GUI grounding.

Results in Table. 13 demonstrate that on OmniAct-Desktop, our model outperforms SeeClick in grounding-related operations, resulting in a higher overall Action Score. Additionally, the label-marking model of SoM struggles to accurately identify interactive elements from desktop screen images, leading to poor performance on this task, consistent with findings in (Kapoor et al., 2024).

| Method | Grd.Acc | Action Score |
|---|---|---|
| SoM (Kapoor et al., 2024) | 12.3 | 17.6 |
| SeeClick (Cheng et al., 2024) | 55.2 | 23.8 |
| MLLM ○ AGG (Ours) | **62.4** | **27.2** |

Table 13: Results on OmniAct-Desktop split.

**Error Analysis.** We conduct error analysis for our proposed model from grounding task and agent-based task.

*Errors in Grounding Task.* We manually inspected test cases from ScreenSpot and found that a significant portion of failures stem from grounding errors involving unseen buttons or icons on mobile or desktop platforms. Despite being pretrained on extensive web GUI data, our model struggles to localize icons with specific meanings and shapes unique to particular platforms. For example, in Figure. 8 (A-C), the top-left simple-designed icon denotes "save the file", "AC" in a mobile calculator app means "All clear calculator", and the bug-shaped vector icon in the IDE means "debug". In these instances, our model incorrectly predicts the "File" item in the PPT navigation bar and nonsensical positions in cases (B) and (C). Additionally, the model struggles with ambiguous queries, as seen in Figure. 8 (D). From this error analysis, we conclude that capturing specific semantics in long-tail categories, such as platform-specific icons and buttons, remains an open challenge for grounding.

*Errors in Agent-based Tasks.* By inspecting test cases from the Mind2Web (Deng et al., 2024) and OmniAct (Kapoor et al., 2024) benchmarks, we have identified several error patterns in the current model. First, when an MLLM or planner generates incorrect actions or context, it can adversely affect the grounding process. For example, if an MLLM incorrectly outputs the action "type" and subsequently the grounding token, the decoded position is likely to be a search bar or text box,

correlating with the erroneous action. Thus, incorrect context from the MLLM negatively impacts subsequent grounding. Second, similar to errors observed in grounding tasks, element grounding errors in agent-based tasks are more likely when the target element is small and less common in the training set. This analysis underscores the importance of enhancing the model's grounding capabilities, particularly for long-tail categories.

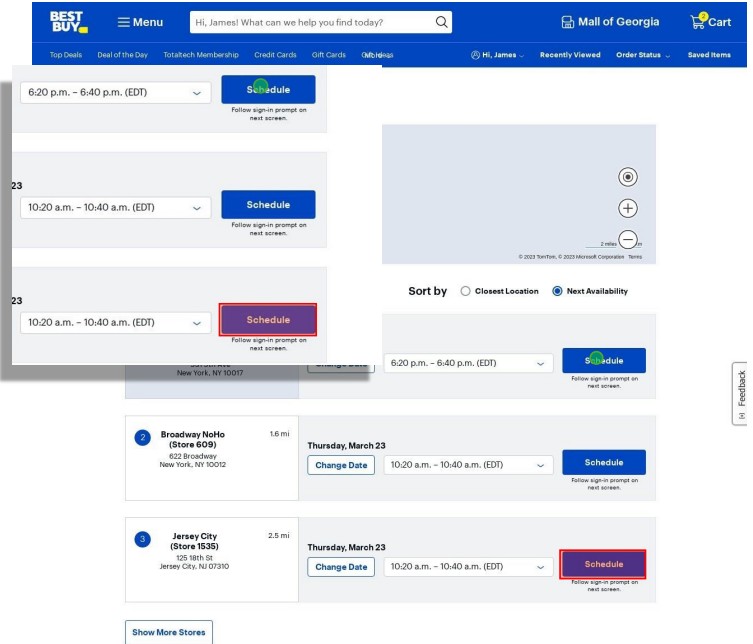

(A) Schedule a repair service for my gaming console closest to the zip code 10001 on March 24, anytime after 6 pm.

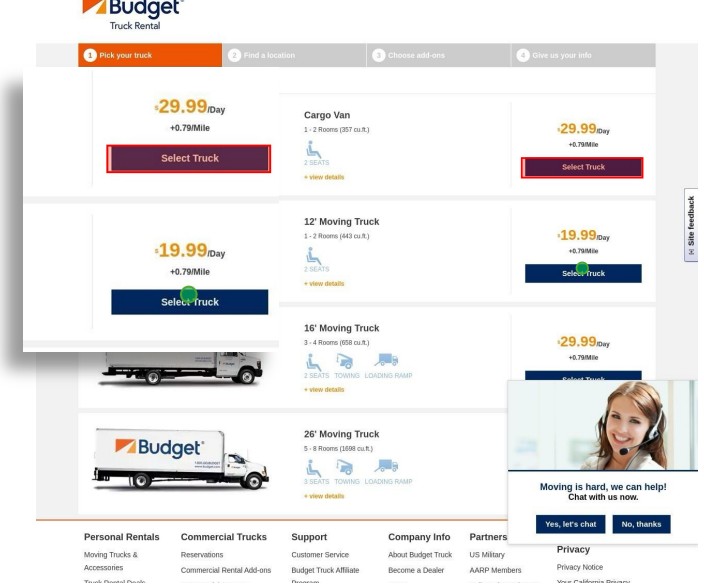

(B) Rent a truck with the lowest rent with two dozen furniture pads for 100 miles at zip 08817 on April 12 at 2:30 pm rented truck is to be returned to the exact location and date, and the pickup and drop off will be at the nearest location.

Figure 7: Examples for comparing our method with text-involved methods.

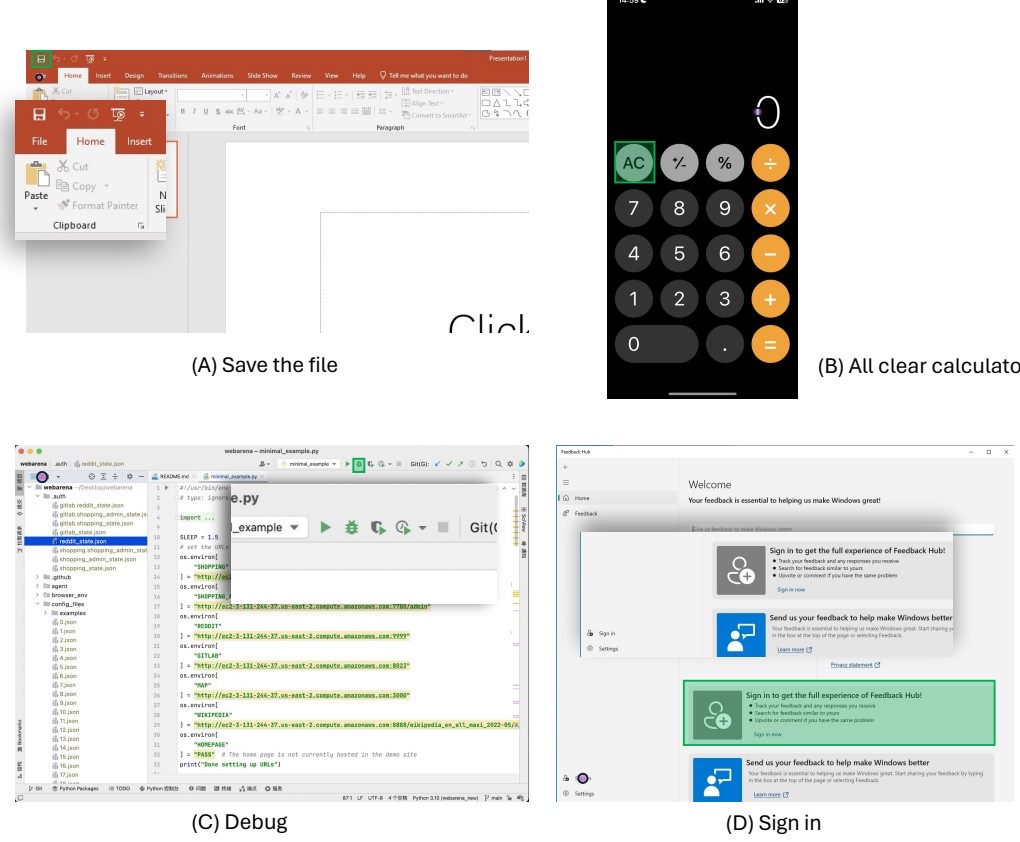

Figure 8: **Visualization of grounding error on ScreenSpot.** The green box indicates the ground truth area. The purple dot is the predicted position of our model. We show four examples with corresponding query.

