# OpenReview forum: "Grounding Multimodal Large Language Model in GUI World"
_ICLR.cc/2025/Conference — ICLR 2025 Poster_

### Official Review · Reviewer_UKHa · 2024-10-19

**Soundness:** 3
**Presentation:** 3
**Contribution:** 3
**Rating:** 6
**Confidence:** 5

**Summary:**

This paper seeks to enhance the grounding capabilities of MLLM-based GUI agents by introducing a new grounding dataset. The authors propose a lightweight grounding plugin trained on this dataset, demonstrating its effectiveness. Additionally, they fine-tune an MLLM-based GUI agent using Qwen-VL, showing promising results on various GUI grounding and navigation benchmarks.

**Strengths:**

- **Comprehensive GUI Grounding Dataset:** The paper presents an automated data collection engine that generates large-scale, diverse, and annotated GUI screenshots. This helps mitigate the shortage of publicly available GUI datasets, significantly contributing to the field.
- **Improved Grounding Capabilities:** The proposed GUI grounding module demonstrates enhanced task accuracy and adaptability, yielding impressive performance across various GUI benchmarks.
- **Robust Technical Contribution:** The paper includes extensive experiments that effectively demonstrate the efficacy of their model for GUI grounding and agent tasks.

**Weaknesses:**

- **Presentation Issues:** The paper appears rushed, with room for improvement in formatting and clarity. For instance, the overview figure on the first page should be placed above the Abstract section instead of between the section title and content of the Introduction. Additionally, there are several missing spaces before `\cite`, and incorrect usage of `\citep` and `\citet`.
- **Overclaimed Contributions and Misused Terminology:** While the ablation studies are appreciated, the authors overstate their contributions. The claim of strong performance in general GUI grounding is unsupported, as they evaluate their model only in web-based tasks (Android-AITW and Mind2Web). Additional experiments on software- or OS-level benchmarks, such as OS-World [1], WindowsAgentArena [2], or VisualAgentBench [3], are necessary to substantiate their claims. Moreover, the term "general GUI grounding" is misleading, as their dataset is collected from webpage. It would be more accurate to call it a "web grounding dataset," although I admit that GUI elements may transfer across different platforms.
- **Writing Suggestions:** The Experiment section lacks clarity, particularly in its structure. It is recommended to introduce details about benchmarks and evaluation metrics in the Experiment Setup section, rather than in the Results section. For instance, information about the sample size and metrics for AITW is provided in line 432 of the Results section but should be included in the setup. A clearer organization would be to separate experiments by benchmarks, as each uses different baselines and metrics.
- **Failure Cases:** Including failure cases of the proposed grounding model and agent would offer a more comprehensive evaluation of the approach and provide insights into its limitations.
- **Dataset Analysis:** While the paper discusses the dataset size, a more detailed analysis is essential. Providing statistics on different levels of data and characteristics of the dataset would offer more insight. Additionally, the authors should address whether the dataset contains NSFW content, which would be crucial for future research and deployment. Moreover, given that your dataset is crawl from the internet, copyright problems should be discussed in this paper.
- **Comparison with Stronger Baselines:** The paper lacks comparison with recently released baselines, such as SeeClick-V. Including such comparisons, especially highlighting cases where their model succeeds while others fail, would further emphasize the model's strengths. Additionally, comparing against Set-of-Marks, a common baseline for non-GUI-specific models, would provide a broader perspective on the effectiveness of their GUI-specific model.

[1] OSWorld: Benchmarking Multimodal Agents for Open-Ended Tasks in Real Computer Environments

[2] WindowsAgentArena: Evaluating Multi-Modal OS Agents at Scale

[3] VisualAgentBench: Towards Large Multimodal Models as Visual Foundation Agents

**Questions:**

- Improve the paper's presentation by addressing formatting issues and clarifying the organization of sections.
- Conduct experiments on additional benchmarks to solidify claims about the model's general GUI grounding abilities.
- Include a more thorough analysis of the dataset and address potential concerns regarding NSFW content and copyright.
- Include failure case analysis.
- Compare with stronger baselines, such as SeeClick-V and Set-of-Marks, to underscore the model's performance relative to current approaches.

**I will consider raising the overall score if author address my concerns, especially the writing and presentation.**

**Details Of Ethics Concerns:**

Copyright problem and concerns of NSFW content as I mentioned in Weakness section.

---

### Official Review · Reviewer_ivbR · 2024-11-04

**Soundness:** 3
**Presentation:** 3
**Contribution:** 3
**Rating:** 6
**Confidence:** 5

**Summary:**

This presents an automated data synthesis for web screenshots and a lightweight model for precise UI element localization, which can be further integrated with MLLMs. The approach enhances MLLM accuracy and adaptability across GUI grounding and agent benchmarks.

**Strengths:**

The paper introduces a lightweight model architecture based on SAM and BERT that achieves strong grounding performance (on ScreenSpot), demonstrating impressive results despite being much smaller (0.4B) than contemporary 7B models.

The writing is clear and cohesive, and the experimental results, including ablation studies, reasonably support both the model design and the data’s effectiveness.

**Weaknesses:**

The web-based data collection pipeline is not novel or technical.

As shown in Table 2, the model struggles with grounding icon elements, which are both essential and challenging in GUI tasks. In contrast, text elements are relatively easier to ground through OCR. Maybe the authors should add more discussions about the discrepancy and the ways to address the challenges.

The model's results do not significantly outperform SeeClick on benchmarks like ScreenSpot, AITW, and MiniWob, despite using a larger and higher-quality training dataset. The significant improvement on Mind2Web is likely due to overlap witt websites and domains (e.g., shopping, travel) rather than a generalized performance boost, whereas SeeClick’s data is more randomly sampled from Common Crawl.

**Questions:**

Overlap Issue in Mind2Web Evaluation: Is the model's high performance on Mind2Web genuinely indicative of its capabilities, or is it influenced by an overlap with similar website domains (e.g., shopping, travel) in the training data?

For the e2e agent setting, despite the ablation study, finally we probably can solve the problem with only an end2end MLLM, without AGG (for example, using Qwen 2 VL, which is known for being capable of grounding). Can you discuss more about the advantage of this work compared to a better finetuned MLLM?

A minor issue: In Fig. 3, does recording "Click at (coordinates)" as part of action history make sense for web agents? (When there is no screenshot recorded for understanding clicking at a point)

---

### Official Review · Reviewer_3qqW · 2024-11-06

**Soundness:** 3
**Presentation:** 3
**Contribution:** 3
**Rating:** 6
**Confidence:** 4

**Summary:**

This paper seeks to enhance the grounding capabilities of MLLM-based GUI agents by introducing a new grounding dataset. The authors propose a lightweight grounding plugin trained on this dataset, demonstrating its effectiveness. Additionally, they fine-tune an MLLM-based GUI agent using Qwen-VL, showing promising results on various GUI grounding and navigation benchmarks.

**Strengths:**

- Comprehensive GUI Grounding Dataset: The paper presents an automated data collection engine that generates large-scale, diverse, and annotated GUI screenshots. This helps mitigate the shortage of publicly available GUI datasets, significantly contributing to the field.
- Improved Grounding Capabilities: The proposed GUI grounding module demonstrates enhanced task accuracy and adaptability, yielding impressive performance across various GUI benchmarks.
- Robust Technical Contribution: The paper includes extensive experiments that effectively demonstrate the efficacy of their model for GUI grounding and agent tasks.

**Weaknesses:**

- Over-claimed Contributions and Misused Terminology: While the ablation studies are appreciated, the authors overstate their contributions. The claim of strong performance in general GUI grounding is unsupported, as they evaluate their model only in web-based tasks (Android-AITW and Mind2Web). Additional experiments on software- or OS-level benchmarks, such as OS-World [1], WindowsAgentArena [2], or VisualAgentBench [3], are necessary to substantiate their claims. Moreover, the term "general GUI grounding" is misleading, as their dataset is collected from webpage. It would be more accurate to call it a "web grounding dataset", although I admit that GUI elements may transfer across different platforms.

- Writing Suggestions: The Experiment section lacks clarity, particularly in its structure. It is recommended to introduce details about benchmarks and evaluation metrics in the Experiment Setup section, rather than in the Results section. For instance, information about the sample size and metrics for AITW is provided in line 432 of the Results section but should be included in the setup. A clearer organization would be to separate experiments by benchmarks, as each uses different baselines and metrics.

- Failure Cases: Including failure cases of the proposed grounding model and agent would offer a more comprehensive evaluation of the approach and provide insights into its limitations.

- Dataset Analysis: While the paper discusses the dataset size, a more detailed analysis is essential. Providing statistics on different levels of data and characteristics of the dataset would offer more insight. Additionally, the authors should address whether the dataset contains NSFW content, which would be crucial for future research and deployment. Moreover, given that your dataset is crawl from the internet, copyright problems should be discussed in this paper.

- Comparison with Stronger Baselines: The paper lacks comparison with recently released baselines, such as SeeClick-V. Including such comparisons, especially highlighting cases where their model succeeds while others fail, would further emphasize the model's strengths. Additionally, comparing against Set-of-Marks, a common baseline for non-GUI-specific models, would provide a broader perspective on the effectiveness of their GUI-specific model.

[1] OSWorld: Benchmarking Multimodal Agents for Open-Ended Tasks in Real Computer Environments

[2] WindowsAgentArena: Evaluating Multi-Modal OS Agents at Scale

[3] VisualAgentBench: Towards Large Multimodal Models as Visual Foundation Agents

**Questions:**

- Conduct experiments on additional benchmarks to solidify claims about the model's general GUI grounding abilities.
- Include a more thorough analysis of the dataset and address potential concerns regarding NSFW content and copyright.
- Include failure case analysis.
- Compare with stronger baselines, such as SeeClick-V and Set-of-Marks, to underscore the model's performance relative to current approaches.

**Details Of Ethics Concerns:**

Copyright problem and concerns of NSFW content as I mentioned in Weakness section.

---

### Official Review · Reviewer_9768 · 2024-11-09

**Soundness:** 3
**Presentation:** 2
**Contribution:** 2
**Rating:** 6
**Confidence:** 5

**Summary:**

In this work the authors propose a framework for improving the GUI grounding of multimodal LLMs, i.e., the localization of relevant elements in a UI from a textual description of the elements.

The framework presented includes an automated way of collecting large scale data across a variety of websites along with automatic annotation of the locations, properties and descriptions of the various UI elements. The authors then clean this data programmatically to remove data with low-quality annotations (mislabeling, missing annotations) to arrive at a clean dataset.

This data is used to train a novel lightweight grounding model to identify the location of a UI element given a textual query. This grounding model achieves SoTA (or close to SoTA) performance on grounding tasks like Screenspot. The authors then present how to integrate their grounding model with a multimodal LLM for completing automation tasks on datasets like MiniWob, Mind2Web and Android in the Wild. They fine-tune an MLLM to work with their grounding model and demonstrate strong performance compared to other image based approaches. Finally, the authors perform ablations to measure the impact of their dataset and model on the performance of their overall framework.

**Strengths:**

1. SoTA image only performance: This work achieves SoTA performance on image based grounding across Screenspot and Automation tasks like MiniWob, Android In the Wild, Mind2Web. Given the recent interest in developing MLLM based GUI agents, and also given that grounding of actions in UIs is identified as a bottleneck for current approaches, this work can be very impactful for the research community.

2. Light-weight model for grounding: Propose a light-weight model for grounding compared to other approaches like Fuyu, SeeClick although this is a single-task model specifically fine-tune for grounding. The authors also presented detailed ablations for the modeling choices which is very informative.

3. Extensible dataset collection approach: The authors present an approach for large scale data collection and annotation which will be of interest to the research community. The dataset seems to result in SoTA performance on grounding tasks.

**Weaknesses:**

1. The paper achieves state of the art results for image based models on grounding/automation tasks. However, I have some questions about the novelty of this work. The dataset collected has many similarities to the dataset from SeeClick with the main difference being the source of the annotations: UI Automation framework vs attributes from the DOM. The model architecture is quite similar to LISA and SAM with detection heads (and loss) replacing segmentation ones.

2. The process for cleaning the annotations is missing some details. In particular, to identify elements with erroneous element names, restructuring the hierarchy of UI elements it is not clear the source of the “correct information” to identify errors and to do restructuring.

3. Error analysis: In lines 484-487 the authors note that image only approaches fall short of text-based approaches. It will be very useful if the authors can analyze the errors compared to a text-based approach and try to identify patterns (if any) which can point to future directions of improvement.

**Questions:**

* "Next, we clean the element data by removing entries with erroneous or missing element names." [203-205]
How are erroneous element names identified? What is the groundtruth to which they are compared to identify a name as erroneous?

* "We also address cases where the element hierarchy is ambiguous or misrepresented by restructuring it to accurately reflect the parent-child relationships between GUI elements." [206-208]
Can you explain how you identify ambiguous or misrepresented hierarchies? Also how do you identify the parent-child relationships between GUI elements?

* "First, we eliminate websites that are either empty or inaccessible as many URLs may lead to pages with no meaningful content, broken links, or require authentication that cannot be handled automatically." [201-203]
Can you specify how many of unique websites are present in the filtered dataset? Also, did this result in any of the topics listed in Table 10 being completely removed?

* Did you conduct any ablations for the filtering and cleaning of the pre-training data on Screenspot or other evals? Any numbers here will be useful for differentiating the pre-training dataset from other existing datasets.

* For the agent/UI automation evals, are the models separately fine-tuned for each of Miniwob, AiTW and Mind2Web? Do you have a comparison to one model trained on all 3 datasets? It will be useful to assess the robustness of the grounding model across different UI types.

* For mind2web, did you run any evals that can be directly compared to a text-based multiple choice or Set of Marks setting (in terms of using the same LLM output for planning)? I think it will be useful to see what is the gap between different grounding methods since you have SoTA image based grounding.

* Can you specify what is the metric/dataset used for Tables 8 and 9?

* For mind2web, can you verify if some of the test-set domains are present in your grounding training set? I don't think it is a big problem but it will be useful to know for the test-website and test-domain splits.

* Do you have any experiments on the scaling of grounding performance with more automatically generated training data? This could indicate if there is any headroom with scaling of pre-training data.

---

### Meta-Review · Area_Chair_ENQC · 2024-12-20

**Metareview:**

The reviewer recommendation is quite unanimous for the acceptance of the paper. On the positive side, the reviewers recognized the importance of the data pipeline that the work contributed. The automated data collection engine generates a large-scale, diverse, and richly annotated dataset of web GUI screenshots, addressing the scarcity of publicly available GUI datasets. The proposed grounding model demonstrates enhanced accuracy and adaptability across different GUI platforms, leading to impressive performance on various benchmarks. The reviewers also liked the extensive experiments and ablation studies conducted in the work, which effectively support the model design and demonstrate the efficacy of both the dataset and the model for GUI grounding and agent tasks.

On the negative side, the reviewers questioned the limited generalizability: While the model shows strong performance on web-based GUI tasks, its generalizability to other GUI platforms needs further evaluation on benchmarks like OS-World, WindowsAgentArena, or VisualAgentBench. The authors seem to also overclaim their contribution and misused the terminologies. There are also a number of minor presentation issues.

In conclusion, despite some initial weaknesses, the authors have effectively addressed most of the reviewers' concerns during the rebuttal phase. The paper makes a valuable contribution by introducing a comprehensive GUI grounding dataset and a lightweight yet effective grounding model. The extensive experiments and ablation studies convincingly demonstrate the model's capabilities and the impact of the dataset. While further evaluation on diverse GUI platforms is encouraged, the current work shows promising results and has the potential to advance research in MLLM-based GUI agents.

**Additional Comments On Reviewer Discussion:**

The authors have diligently addressed the reviewers' concerns, providing clarifications, additional experiments, and analyses to support their claims and improve the paper. The reviewers are overall happy with the rebuttal, and Reviewer UKHa has increased the score.

---

### Decision · Program_Chairs · 2025-01-22

Accept (Poster)